# Chemical Vapor Deposited Mixed Metal Halide Perovskite Thin Films

**DOI:** 10.3390/ma14133526

**Published:** 2021-06-24

**Authors:** Siphesihle Siphamandla Magubane, Christopher Joseph Arendse, Siphelo Ngqoloda, Franscious Cummings, Christopher Mtshali, Amogelang Sylvester Bolokang

**Affiliations:** 1Department of Physics and Astronomy, University of the Western Cape, Private Bag X17, Bellville 7535, South Africa; sngqoloda@uwc.ac.za (S.N.); fcummings@uwc.ac.za (F.C.); sbolokang@csir.co.za (A.S.B.); 2iThemba LABS, National Research Foundation, Somerset West 7129, South Africa; mtshali@tlabs.ac.za; 3CSIR Material Science & Manufacturing, Advanced Materials and Engineering, Meiring Naude Road, Pretoria 0001, South Africa

**Keywords:** mixed metal-mixed halide perovskites, chemical vapour deposition (CVD), elemental stoichiometry, structure-property relationship

## Abstract

In this article, we used a two-step chemical vapor deposition (CVD) method to synthesize methylammonium lead-tin triiodide perovskite films, MAPb_1−x_Sn_x_I_3_, with x varying from 0 to 1. We successfully controlled the concentration of Sn in the perovskite films and used Rutherford backscattering spectroscopy (RBS) to quantify the composition of the precursor films for conversion into perovskite films. According to the RBS results, increasing the SnCl_2_ source amount in the reaction chamber translate into an increase in Sn concentration in the films. The crystal structure and the optical properties of perovskite films were examined by X-ray diffraction (XRD) and UV-Vis spectrometry. All the perovskite films depicted similar XRD patterns corresponding to a tetragonal structure with I4cm space group despite the precursor films having different crystal structures. The increasing concentration of Sn in the perovskite films linearly decreased the unit volume from about 988.4 Å^3^ for MAPbI_3_ to about 983.3 Å^3^ for MAPb_0_._39_Sn_0_._61_I_3_, which consequently influenced the optical properties of the films manifested by the decrease in energy bandgap (E_g_) and an increase in the disorder in the band gap. The SEM micrographs depicted improvements in the grain size (0.3–1 µm) and surface coverage of the perovskite films compared with the precursor films.

## 1. Introduction

Three-dimensional (3-D) organo-lead halide perovskites, which form a perovskite structure of ABX_3_, have received tremendous research interest over the past decade [1]. As a result, we have seen tremendous progress in the solar cell (SC) performance from 3.8% to 25.5% power conversion efficiency (PCE) of these lead-based perovskite SCs. This is due to their high optical absorption coefficients, high charge carrier mobilities, long charge carrier diffusion lengths, low exciton binding energy, bandgap tunability, and low-temperature solution processability [2,3,4,5,6,7,8,9,10]. Despite this overwhelming development, considering the nature of the solar absorption spectrum, the maximum theoretical efficiency can be achieved in a single junction device with an optimum energy bandgap (E_g_) of about 1.1 to 1.3 eV [11]. Broadening the range of solar absorption to the near-infrared (IR) region could further enhance the performance of perovskite SCs [12]. Therefore, the relatively wide E_g_ (1.5–2.2 eV) of organo-lead halide perovskite dramatically limits the sensitivity of the film in the IR region of the solar spectrum [13]. Another major drawback that is associated with lead-based perovskites is the toxicity of the lead (Pb) element and its compounds, which pose threats to the environment and human well-being, thus restricting the commercialization of Pb-based perovskite solar cells (PSCs) [7,14,15,16,17,18].

The challenges that are associated with Pb-based perovskites have sparked research interest in the partial and complete replacement of Pb with tin (Sn) in the perovskite structure, not only to reduce the toxic Pb but also to extend the absorption range to above 960 nm wavelengths, i.e., below 1.3 eV [19,20,21,22,23,24,25]. On one hand, a complete replacement of Pb seems to be improbable at this point due to the low PCE of Sn-based perovskite SCs that is driven by the rapid crystallization of Sn-based perovskite, making the film growth control to be a challenge [26,27,28]. Moreover, Sn-perovskite films are very unstable compared to pure Pb-based perovskites [24,25,27]. This instability is due to the easy oxidation of Sn^2+^ to Sn^4+^ that causes Sn vacancies in the film, thus promoting non-radiative recombination, which leads to reduced charge carrier lifetimes in the PSCs [29,30]. On the other hand, tremendous efficiency improvements and various discoveries have been made in the low-bandgap (1.17–1.1.38 eV) Pb-Sn PSCs [31,32,33,34].

The technique that is mostly employed to deposit the low-bandgap mixed Pb-Sn halide perovskite is one-step spin-coating with antisolvent treatment followed by annealing at about 100 °C for crystallization [31]. Different ratios of solvents, i.e., Dimethylformamide (DMF) and Dimethyl sulfoxide (DMSO), have been explored and different organic cations such as methylammonium (MA), formamidinium (FA), cesium (Cs), and halides like iodine (I), bromine (Br), chlorine (Cl), or a combination of these halides have been investigated [30,35,36,37,38,39,40,41,42,43]. It has been observed that uniform, large-grained, and pinhole-free films with high surface coverage are requirements for the high performance of Sn-based PSCs [31]. However, there seems to be a consensus that the control of crystal growth becomes complicated as the proportion of Sn increases in the perovskite structure [32,44,45]. The crystal structure of methylammonium lead-tin triiodide (MAPb_1−x_Sn_x_I_3_) perovskites transition from tetragonal phase (space group *14cm*) to pseudo-cubic phase (space group *P4mm*) when *x* nears 0.5 (for *x* ranging from 0 to 1) [31]. During this transition the valence band (VB) and the conduction band (CB) undergo an energy shift from −5.45 and −3.90 eV (for *x* = 0) to −4.73 and −3.63 eV (for *x* = 1), respectively [28,31,33,46,47,48]. These changes are attributed to the perovskite crystal lattice compression and distortion as a result of increased Sn incorporation [49,50,51,52].

Additional efforts, such as the incorporation of additives like halogens and thiocyanate (SCN) series in the perovskite structure, film surface passivation, and interface engineering aiming at reducing grain boundaries and charge scattering thereby extending the charge carrier lifetimes, and increasing the charge diffusion lengths, have all been explored. Halide doping, including Cl or Br, is another effective strategy to enhance the film quality of the mixed Pb-Sn perovskites [49,53]. This allows the precursor solution to convert to a perovskite consisting of intermixing of Br/Cl and iodine in the lattice, thereby reducing Sn and iodine vacancies in the film [49,53]. Other researchers have proved that similarly to halide doping, the addition of antioxidants and ascorbic acids such as tin fluoride (SnF_2_), tin sulfide (SnS), quaternary ammonium halide (Me_4_NBr), tetramethylammonium iodide (Me_4_NI), methylammonium bromide (MABr), lead thiocyanate (Pb(SCN)_2_), and other thiocyanate (SCN) salts can improve the perovskite film quality [31]. They ascribed this to reductive additives that do not only prevent Sn^2+^ oxidation, but also promote the crystallization growth through strong coordination between PbI_2_/SnI_2_ and an organic cation or a combination of organic cations which allows the complete interaction and conversion of metal halides into perovskite [53,54,55,56,57,58,59,60,61].

Attempts to improve the charge diffusion length, including but not limited to the incorporation of cadmium iodide (CdI) and/or metallic Sn in the low bandgap Pb-Sn perovskite, have also been explored [30,59,60]. This reduces the free hole concentration and electron density, which leads to an improved diffusion length of up to 3 μm [31]. Other attempts, such as the deposition of ultrathin bulk-heterojunction (BHJ) organic semiconductor layer as an intermediary between the hole transport layer (HTL) and the mixed low bandgap perovskite to minimize the energy loss by reducing energy level mismatch and passivating defects in the HTL/perovskite interface have been reported [58,61,62,63].

The need to produce stable perovskite films that will withstand exposure to outdoor conditions without introducing morphological and structural changes that consequently affect the electronic properties of the films, remains. This article, therefore, aims to optimize and establish the processing conditions to synthesize uniform and stable low bandgap mixed Pb-Sn perovskite films that exhibit homogenous elemental distribution using a chemical vapor deposition (CVD) technique. We have successfully controlled the concentration of Sn in the perovskite structure and then examined the structure-property relationship of these perovskite films in relation to the deposition technique. The technique is divided into two steps that include (i) the co-evaporation and deposition of tin chloride-lead iodide (SnCl_2_-PbI_2_) compound films, followed by (ii) the conversion of these compound films into perovskite through exposure to methylammonium iodide (MAI) vapor [64]. The two-step deposition technique is employed because it promises a better understanding of the formation mechanism of the perovskite films through controlling the precursor films deposition parameters and the conversion parameters, independently [62,63]. Also, the CVD method avoids the use of solvents to dissolve the precursor chemicals, which results in highly stable, uniform, largely grained, and pin-hole free perovskite films [62,63]. This is due to the low pressure (1 mbar) and inert environment in which the processing/deposition takes place. In terms of large-scale production and reproducibility, the CVD technique is the strongest contender over the cost-efficient solution-processed perovskites, because it is well established in the semiconductor industry especially in silicon-based research [64,65].

## 2. Materials and Methods

### 2.1. Precursor Thin Films Deposition

Mixed SnCl_2_-PbI_2_ precursor films were deposited on 1.5 × 2 cm^2^ corning glass substrates that were cleaned via sonication in boiling hellmanex solution and isopropanol for 15 min each in an ultrasonic bath. After that, the substrates were rinsed in boiling deionized (DI) water and then dried with a nitrogen (N_2_) gun. The deposition system that was used is composed of a horizontal quartz tube enclosed with the three-independent temperature-controlled zone furnace (Brother XD 1600MT Furnace Co., LTD, Zhengzhou, China) [64]. All the materials were purchased from Sigma Aldrich (St. Louis, MO, USA) and were used as obtained without any further purification. A ceramic boat containing a mixture of different powder amounts of 98% purity SnCl_2_ (i.e., 2.5, 5, 10, and 15 mg) and 20 mg of 99.9% PbI_2_ was placed in the 1st zone while the substrates were placed in the 2nd zone at about 16–20 cm away from the source boat [64]. The tube of the furnace was then sealed and pumped down to a base pressure of 0.04 mbar. Next, the tube was purged with N_2_ gas at a rate of 200 sccm (i.e., standard cubic centimeters per minute) to flush away all form of contaminations pre-existed in the tube. Then, the source boat was heated to a temperature of 380 °C while flowing N_2_ gas at a rate of 500 sccm to direct the vapor towards the substrates. The deposition pressure was between 2.5 to 3 mbar and the duration was set for 20 min, after which the sources were evaporated completely. For comparison purposes, pure PbI_2_ and SnCl_2_ films were deposited in a similar manner except that the evaporation temperature was 270 °C for the SnCl_2_ source and 380 °C for PbI_2_.

### 2.2. Precursor Films Conversion into Perovskites

The as-deposited precursor films were converted into perovskite films through exposure to MAI (CH_3_NH_3_I) vapor in the same CVD system [64]. A ceramic boat containing 60 mg of MAI (Dyesol) was placed at the center of zone 1 while the precursor films, i.e., PbI_2_, SnCl_2_, and mixed SnCl_2_-PbI_2_ compound films were placed in the second zone, downstream, at about 18 cm away from the source [64]. The tube of the furnace was then sealed and pumped down to also a base pressure of 0.04 mbar. Next, the tube was purged with N_2_ gas at a rate of 200 sccm. Then, the MAI was heated to a temperature of 180 °C while the precursor films were heated and maintained at about 85 °C while a 100 sccm flow rate of N_2_ gas was maintained to direct the MAI vapor towards the precursor films. The conversion pressure was maintained at about 0.9 mbar and the duration was set for 50 min.

### 2.3. Characterisation Methods

A Zeiss Auriga (Carl Zeiss Jena GmbH, Jena, Germany) field emission gun scanning electron microscope (FEG-SEM) equipped with an energy dispersive X-ray detector, was operated at an acceleration voltage of 5 kV to probe the surface morphology and the elemental composition of the precursor films. The thickness and the roughness of the films were measured using a Dektak 6M stylus profilometer (Veeco Instruments, Inc., Tucson, AZ, USA). The depth profile and the composition of the films were further examined through Rutherford Backscattering Spectrometry (RBS) (Cape Town, South Africa) using an ion emanating from 3 MV Tandetron accelerator (High Voltage Engineering Europa BV, Amersfoort, Netherlands) hosted at iThemba Laboratory for Accelerator-Based Science (Cape Town, South Africa). A 3.05 MeV He^++^ ion beam was used, where the average current of 60 nA and the total collected charge of 40 µC were maintained for all samples. A Bruker AXS D8 diffractometer (Bruker, Karlsruhe, Germany) with an irradiation line Ka1 of copper (kCuKa1 = 1.5406 Å) operating at a voltage of 40 kV and a current of 35 mA was used for the investigation of the crystal structure and phase composition of the films. During this experiment, the samples were scanned over a 2θ-range of 10° to 100°, with a step size of 0.026°.

## 3. Results and Discussions

### 3.1. Mixed SnCl_2_-PbI_2_ Precursor Films

#### 3.1.1. Surface Morphology

Throughout the discussion of the precursor films, samples will be referred to according to the SnCl_2_: PbI_2_ mass ratio used during the preparation of these precursor films. It is observed from the SEM micrographs in Figure 1a that PbI_2_ form a highly compact film composed of large grains that are tightly packed across the whole substrate. This behavior is consistent with that observed from the literature [66,67,68]. From Figure 1b, it is observed that SnCl_2_ forms randomly shaped, distinguishable islands with clearly defined boundaries. Furthermore, the SnCl_2_ forms clusters or agglomerates that are circled in red in Figure 1b. This SEM micrograph is different from those obtained by R. Felix et al. for SnCl_2_ films that were deposited using a 4-pocket evaporator attached to the ultra-high vacuum (UHV) deposition chamber [69]. Their results suggested that the growth mechanism of the SnCl_2_ film is divided into two-part; the lateral formation of the seeding layer followed by the vertical growth of the SnCl_2_ islands [69]. This suggests that our film might have reacted with moisture to form impurities such as tin oxide or hydrogen chloride since their films were prepared and studied in the system without exposure to ambient air conditions.

Figure 1c,d shows the SEM micrographs of mixed PbI_2_-SnCl_2_ compound films composed of different amount of SnCl_2_ (i.e., 2.5 mg, 5 mg, 10 mg, and 15 mg mixed with 20 mg of PbI_2_). Overall, it is observed that the presence of PbI_2_ improves the quality of the film of SnCl_2_. Figure 1c ((2.5:20) mg) depicts a film composed of clearly defined grains with grain sizes in the range between 200–400 nm and a high density of small-sized pores. Increasing the amount of SnCl_2_ to 5 mg in the film results in the coalescing of the small grains thereby forming larger grains. This sample still possesses low surface coverage due to the increase in the size of pores (Figure 1d). Figure 1e,f show that further increasing the amount of SnCl_2_ in the mixed metal-halide compound results in the deterioration of the quality of the films, hence, the observed disordered films with a larger number of pores that extend through the film to the substrate, which resulted in poor surface coverage. The thickness evolution of the compound films upon the increase of SnCl_2_ amount was measured using Dektak profilometer and plotted in Appendix A. It is observed from Appendix A that the thickness of the films increases linearly as the amount of SnCl_2_ that was added in the crucible increases.

#### 3.1.2. Rutherford Backscattering Spectrometry (RBS)

RBS was used for the determination of the stoichiometry and the depth profiles of the films. Figure 2a shows the experimental RBS spectra of PbI_2_ and mixed SnCl_2_-PbI_2_ compound thin films. Energy X-ray dispersive spectroscopy (EDS) data, not shown, was used for initial composition as required by SIMNRA software to begin the fitting process; with the simulated RBS spectra provided in Appendix A. The PbI_2_ film (black) shows two peaks of alpha particles backscattered from Pb and I atoms with surface energies of 2826.9 keV and 2694.2 keV, respectively. Although this is a single-layered film, the Pb peak appears at higher energy as compared to the I peak due to the difference in their atomic masses. This spectrum (i.e., PbI_2_ film) could be simulated using two layers with a total thickness and the average stoichiometry of 3.61 × 10^17^ atoms/cm^2^ and Pb_0.35_I_0.65_, respectively. Refer to Table 1 below for layer-by-layer details. The stoichiometry of the PbI_2_ film reported in this work corresponds to that reported by Popov et al. [70] and Tsevas et al. [71].

It can be observed from Figure 2a that mixing SnCl_2_ and PbI_2_ for the formation of compound films results; (i) in the quenching of the Pb peak intensity, (ii) peak broadening toward lower energies of iodine peak because of Sn peak with a surface energy of 2694.2 keV, which is close to the ions scattered from iodine, (iii) the development of Cl peak with a surface energy of 1952.9 keV. The addition of more SnCl_2_ in the mixed SnCl_2_-PbI_2_ compound films results in the progressive quenching of the Pb peak intensity and Sn/I peak broadening, as can be seen from Figure 2a. The quenching of scattering yield from the Pb atoms (especially at lower energies) as the amount of SnCl_2_ is increased implies that there is a spontaneous reaction between SnCl_2_ and PbI_2_ during the deposition, which might have been facilitated by the diffusion of surface atoms toward the substrate and/or the migration of bottom atoms toward the surface of the films. The Sn/I peak broadening and increasing intensity of Cl peaks indicate the progressive increase of Sn and Cl concentrations within the mixed SnCl_2_-PbI_2_ compound films.

Since the concentration of Sn increases upon the addition of SnCl_2_, the quenching of Sn/I peaks suggest that Cl atoms replace some I atoms in the mixed SnCl_2_-PbI_2_ compound films. These mixed SnCl_2_-PbI_2_ compound films that were deposited using different amount of SnCl_2_, i.e., 2.5 mg (red), 5 mg (blue), 10 mg (magenta), and 15 mg (green) mixed with 20 mg of PbI_2_ were simulated using three layers with a total thickness (×10^17^ atoms/cm^2^) and average stoichiometry of 4.22 & Pb_0.24_Sn_0.11_I_0.30_Cl_0.36_, 4.59 & Pb_0.23_Sn_0.21_I_0.11_Cl_0.43_, 4.94 & Pb_0.23_Sn_0.27_I_0.04_Cl_0.45_ and 6.01 & Pb_0.19_Sn_0.30_I_0.02_Cl_0.48_, respectively. Refer to Table 1 for the layer-by-layer details. It is noted from the table that surface layers of the films have more Pb atoms compared to the bottom layers that have more Sn atoms. This suggests that the SnCl_2_-PbI_2_ compound films formation is divided into two; (i) the deposition of the SnCl_2_ layer first followed by (ii) the deposition of the PbI_2_. The films were deposited on a glass substrate, which could be simulated with a thickness and stoichiometry of 3.50 × 10^22^ atoms/cm^2^ and Si_0.19_O_0.72_Al_0.05_Ba_0.04_As_0.003_, which correlates to what was obtained by Kumar et al. [72]. Figure 2b shows the compositional ratios of Sn/(Sn + Pb) and (Cl/Cl + I) deposited on the films. It can be observed from Figure 2b that the increased amount of SnCl_2_ in the source boat translated to the increase in Sn and Cl atoms in the films. A similar trend is portrayed by both Sn and Cl in comparison to Pb and iodine, respectively. This suggests that the separation distance between the source and the substrate is within the average migration distance of the vapors.

#### 3.1.3. Crystal Structure and Phase Composition

X-ray diffraction (XRD) was employed to understand the effect of the SnCl_2_ content on the structure of the SnCl_2_-PbI_2_ compound thin films. Figure 3a shows XRD patterns of PbI_2_ thin film, SnCl_2_ thin film, and SnCl_2_-PbI_2_ compound films as named according to the amount of SnCl_2_ used. On one hand, the PbI_2_ thin-film depicted peaks located at 2θ values of 13.05°, 25.90°, 39.05°, and 52.77° assigned to (001), (011), (110), and (202) lattice planes, respectively. This PbI_2_ film formed a hexagonal structure, *P321* space group (ICSD code: 01-075-0983), with the lattice parameters, *a* and *c*, of 4.606 Å and 6.774 Å, respectively. On the other hand, the SnCl_2_ thin-film depicted peaks located at 2θ values of 14.66°, 21.90°, 29.21°, 36.64°, 52.06°, and 60.14° ascribed to (111), (002), (222), (132), (333), and (062) lattice planes, respectively. It turns out that the film formed is not pure SnCl_2_, rather hydrated tin chloride (i.e., [SnCl_4_(H_2_O)_2_]3H_2_O) film with a monoclinic structure, *C12/c1* space group (COD ID: 1534841), with the lattice parameters, *a, b* and *c*, of 7.760 Å, 9.555 Å, and 4.483 Å, respectively [73].

It is believed that as the SnCl_2_ film was deposited in the chamber, it reacted with and absorbed the ambient moisture upon exposure to the environment during the transfer of the sample. For mixed SnCl_2_-PbI_2_ films, it is observed from Figure 3a that the prominent Bragg diffraction peaks correspond to that of [SnCl_4_(H_2_O)_2_]3H_2_O and Pbl_2_ phases, indicating mixed-phase compounds. Also, low-intensity diffraction peaks unrelated to [SnCl_4_(H_2_O)_2_]3H_2_O) and/or PbI_2_ are observed for the 5, 10, and 15 mg SnCl_2_ samples. These peaks were located at 2θ values of 21.10°, 21.52°, 23.37°, 30.82°, 33.73°, and 34.28° which were ascribed to (200), (011), (201), (211), (013), and (212) lattice planes, respectively. These peaks, which are highlighted in grey in Figure 3a belonged to tin chloride iodide (SnClI) which form an orthorhombic structure, *Pnma* space group (ICSD code: 01-073-0826), with the lattice parameters, *a*, *b*, and *c*, of 8.42 Å, 4.43 Å and 10.04 Å, respectively. Figure 3b shows the (001) and (222) peaks of the SnCl_2_-PbI_2_ compound films, compared to that of PbI_2_ and SnCl_2_, respectively. It is observed in Figure 3b that the intensity of the (222) peak, which corresponds to [SnCl_4_(H_2_O)_2_]3H_2_O, increases as the amount of Sn increases in the film. The decrease in intensity of the (001) peak intensity, which corresponds to the PbI_2_, as the amount of Sn content increases supports the RBS discussion. Also, all the (001) peaks of SnCl_2_-PbI_2_ compound films shift toward lower 2θ angles with increasing SnCl_2_ concentration. This peak shift is due to the change in the chemical composition that could be ascribed to the chemical pressure caused by the change in lattice parameters [74,75]. The overall decrease in the peak intensities as the concentration of Sn increases suggests poor periodicity in the newly formed compound structures.

#### 3.1.4. Optical Properties

The optical absorption spectra of PbI_2_, SnCl_2_, and the mixed SnCl_2_-PbI_2_ compound thin films was recorded as a function of wavelength in the range of (200–750) nm at room temperature, as shown in Figure 4a. The absorption onsets of the SnCl_2_ and PbI_2_ thin films were found to be at about 316 and 512 nm, respectively [66,76,77,78]. Furthermore, SnCl_2_ thin-film depicts an absorption peak centered at about 254 nm, while the PbI_2_ film depicts a characteristic absorption peak centered at about 405 nm, featuring a shoulder at 497 nm [66,77]. The absorption peak centered at about 252 nm is the contribution from the glass substrate on which the films were deposited. Generally, upon mixing PbI_2_ with SnCl_2_, an increase in the optical absorption intensity of the films in the wavelength range between 250–400 nm is observed, ascribed to the increase in the thickness of the films. The most noticeable feature in the optical properties of the mixed SnCl_2_-PbI_2_ compound thin films in comparison to that of pure PbI_2_ films is the red-shift of absorption onset. This necessitated the calculation of the optical band gap of the materials under investigation. The direct allowed band gap transition values were obtained by extrapolating the linear portion of the (*αhv*)^2^ versus *hv* plot, as depicted in Appendix A. The corresponding values of E_g_ were found and plotted in Figure 4b. The E_g_ values increase from 2.42 eV for the PbI_2_ film to 3.31 eV for the mixed SnCl_2_-PbI_2_ compound film (15:20 mg) as the concentration of SnCl_2_ increases. The increase in the E_g_ confirms increasing concentrations of Sn in the films, since the E_g_ of SnCl_2_ is 3.92 eV. This further supports and confirms the RBS and XRD results.

### 3.2. Perovskite Structure-Property Relationships

#### 3.2.1. Surface Morphology

Figure 5 shows the top view SEM images of the Pb-based, Sn-based and Sn-Pb based perovskite films prepared using the precursor films that were discussed in the previous section. Compared with the precursor films in Figure 1, there are improvements in the grain size and surface coverage of the perovskite films. It is clear from the morphology in Figure 5a that MAPbI_3_ film is composed of well-defined, large grains (~1 µm) that are compact across the entire surface. The MASnI_3_ film is also composed of large grains, but with a porous morphology forming agglomerates, Figure 5b, which is due to the disorganized hydrated SnCl_2_ precursor film shown in Figure 1b. In the case of mixed metal-based perovskites, the crystal sizes of the films decrease as compared to the pure metal-based perovskite, Figure 5c,d. There is no significant difference in the surface morphology of the SnCl_2_-PbI_2_ compound films, except for an increased in roughness as the Sn concentration increases.

It is worth noting that the poly-porous morphology that was observed for mixed SnCl_2_-PbI_2_ compound films in Figure 1, improved to become compact perovskite films composed of crystals covering the entire surface. This is attributed to the MAI penetrating through the pores leading to sufficient reaction between mixed metal halides and MAI, thus forming completely converted perovskites. It is confirmed in Appendix A that the thickness and the roughness of the films increase as the concentration of Sn increases. However, a massive increase in the perovskite films is attributed to the conversion process, i.e., the exposure of the precursor films to MAI vapor at a substrate temperature of 85 °C. The thickness increased by a factor of about 1.7 from the precursor films to the perovskite films, which indicates optimum conversion parameters.

#### 3.2.2. Crystal Structure and Optical Properties

The XRD patterns of the perovskite films obtained from PbI_2_, SnCl_2_ and mixed SnCl_2_: PbI_2_ compound films are shown in Figure 6a. All the perovskite films possess similar patterns in terms of peak positions with the exception of the diffraction intensities. This suggests that iodine from the MAI replaces the Cl in SnCl_2_ and mixed SnCl_2_-PbI_2_ compound films [79]. Also, the PbI_2_ stabilizes the mixed metal halide compound films and reduces their susceptibility to moisture compared to the SnCl_2_ film. The peaks located at 2θ values of 14.20°, 24.57°, 28.49°, 31.96°, 40.98°, 43.30°, and 50.29° could be ascribed to (110), (202), (220), (222), (400), (420), and (404) lattice planes, respectively. According to the crystallography open-sources database (COD), these perovskite films form a tetragonal structure with *I4cm* space group (COD ID: 4335638, 4335641, 4335636, for the Pb-based, mixed Sn-Pb based, and Sn-based perovskites, respectively) [80]. The MAPbI_3_ and MAPb_0.69_Sn_0.31_I_3_ show additional peaks at 12.66° (denoted by an asterisk) and 58.97° ascribed to PbI_2_ (001) and MAPbI_3_ (440) lattice plane, respectively. The PbI_2_ peak denotes slightly incomplete conversions. This could have been because of tightly packed crystals in the PbI_2_ and Pb_0.24_Sn_0.11_I_0.30_Cl_0.36_ compound films, which could limit the infusion of MAI during the conversion stage.

The highly intense MAPbI_3_ sample peaks are ascribed to the neat, large, and uniformly distributed crystals, while the MASnI_3_ sample with second-most intense peaks is ascribed only to its large crystals. The lower peak intensities of MAPb_1−x_ Sn_x_I_3_ films denoted poor crystallinity of these perovskite films compared to the MAPbI_3_ film. Figure 6b shows the inverse relation of the unit cell volume with increasing concentration of Sn. The unit cell volume of the MAPbI_3_ perovskite decreases from 988.4 Å^3^ to about 983.3 Å^3^ upon the incorporation of Sn in the structure, which is coherent as the volume of the MASnI_3_ perovskite is 970.3 Å^3^. This is due to the smaller ionic radius of Sn compared to that of Pb. Table 2 shows the lattice constants, FWHM of (110) diffraction peak located at 2θ value of 14.2° and the average crystallite size (D) estimated using Scherrer equation, D = kλ/ꞵcosθ, where k = 0.89, λ = 0.154 nm, and θ = 14.3° are Scherrer’s constant, X-ray wavelength, and Bragg diffraction angle, respectively. The average crystallite size of the pure perovskites, i.e., MAPbI_3_ and MASnI_3_ are 71.9 nm and 65.6 nm, respectively. These values are larger than those obtained for mixed metal perovskites as evident from Table 2. This agrees with the SEM results; considering that the pure perovskites films possessed larger grains compared to those of mixed metal perovskites. The crystallite sizes decreased upon the increasing concentration of Sn in the perovskite films from 71.9 nm for MAPbI_3_ to 50.9 nm for MAPb_0.39_Sn_0.61_I_3_. The crystallite size of MAPb_0.69_Sn_0.31_I_3_ was calculated to be 45.8 nm, which is considered as an outlier as observed in Appendix A.

The UV-Vis absorption spectra of the perovskite films are shown in Figure 7a. The MAPbI_3_ depicts characteristic absorption peaks centered at the wavelength of about 357 nm and 470 nm and a shoulder at 747 nm, whereas the MASnI_3_ perovskite shows absorption extending from 365 to 1000 nm with characteristic absorption peaks centered at about 536 and 850 nm. The absorption peak centered at about 252 nm is the contribution from the glass substrate on which the films were deposited. The increase in the optical absorption intensity upon the addition of Sn in the structure is attributed to the increase in the perovskite films thicknesses. It is noticed from Figure 7a that the absorption band edges of the films are not sharp and well-defined. This produces band tails that extend into the band gap, which increases with increasing Sn concentration that is associated with an increasing defect-concentration in the films. These band tails, also referred to as the Urbach tails, is associated with the Urbach energy (E_u_) that can be extracted using the relation α = α_0_exp(E/E_u_), where α is the absorption coefficient and E the photon energy.

Figure 7b shows the band gap (E_g_) evolution obtained from Tauc plots depicted in Appendix A. The incorporation of Sn decreases the band gap of the films from about 1.59 eV for MAPbI_3_ to about 1.26 eV for MAPb_0.39_Sn_0.61_I_3_ and the lowest E_g_ of 1.15 eV was obtained for MASnI_3_, as depicted in Figure 7b and Table 3. Apart from absorption band-edge shift toward high wavelength, Sn incorporation also retards the structure of MAPbI_3_ slightly, augmented by the diminishing of MAPbI_3_ characteristic absorption features, which corresponds to the results in the literature [31,32,33,67]. However, contrary to recent studies, our results do not show absorption bandgap bowing behavior, which is the reduction of the E_g_ of the mixed metal perovskites to values below the end compounds i.e., MAPbI_3_ (1.59 eV) and MASnI_3_ (1.15 eV) in this case. There is an ongoing debate about the origin and magnitude of the bandgap bowing in mixed metal perovskites; whether it is due to the chemical effects, structural effects, or a combination of both [81,82,83]. Nonetheless, it is believed that the 61% maximum Sn concentration employed in this study is below the bandgap bowing threshold for the deposition system and the parameters used.

The E_u_ values, which equates to the widths of the band tails of the defect states in the band gap and thus the density of disorder in the films, are tabulated in Table 3 and were obtained by plotting the natural logarithm of the absorption coefficient ln (α) as a function of photon energy and then taking the reciprocal of the slope from the linear part of the curve [84,85], as shown in Appendix A. The MAPbI_3_ has an E_u_ of about 70 meV, which remains constant for the MAPb_0.69_Sn_0.31_I_3_ film with the lowest concentration of Sn. Further increasing the concentration of Sn increased the E_u_ to about 280 meV for MAPb_0.39_Sn_0.61_I_3_, and the MASnI_3_ showed the highest E_u_ of about 530 eV. The increasing trend of E_u_ with increasing concentration of Sn was observed by Zhao et al. and Li et al. [13,86]. This is ascribed to the increased oxygen defect centers in the thin films affiliated with the increased oxidation of Sn^2+^ to Sn^4+^ [13,86]. This observation of an increasing E_u_ with increasing Sn concentration confirms the XRD suggestion that the incorporation of excessive amount of Sn in the Pb based perovskites results in poor crystallization and the high-level defect density that is induced by Sn oxidation.

## 4. Conclusions

We synthesized mixed metal halide perovskite thin films by two step-sequential CVD, which included the deposition of mixed SnCl_2_-PbI_2_ compound as the precursor films and subsequently exposing them to MAI for conversion into perovskites. A controlled the concentration of Sn in the perovskite films was demonstrated, as quantified by RBS. The micrographs of these precursor films depicted a poly-porous surface with pores extending deep to the substrate with an increasing concentration of Sn in the film. These pores might have been helpful in the conversion processes in terms of acting as the passage through which MAI infiltrated the films thus leading to complete conversions into perovskites rather than partial conversions. All the perovskite films depicted similar XRD patterns corresponding to a tetragonal structure with *I4cm* space group regardless of the precursor films having different crystal structures which is positive since the aim of the study was to partially replace Pb atoms with Sn without destroying the core of MAPbI_3_ perovskite structure. This resulted in the decrease of the unit volume of the perovskites from about 988.4 Å^3^ for MAPbI_3_ to about 983.3 Å^3^ for MAPb_0.39_Sn_0.61_I_3_ which was attributed to the smaller atomic radius of Sn than that of Pb. Therefore, the inclusion of Sn in the structure did indeed influence the properties of the resulted perovskite films manifested by the decrease in band gap from about 1.59 eV for MAPbI_3_ to 1.26 eV for MAPb_0.39_Sn_0.61_I_3_ and an increased defect density in the band gap that is associated with the increased oxidation of Sn^2+^ to Sn^4+^.

## Figures and Tables

**Figure 1 materials-14-03526-f001:**
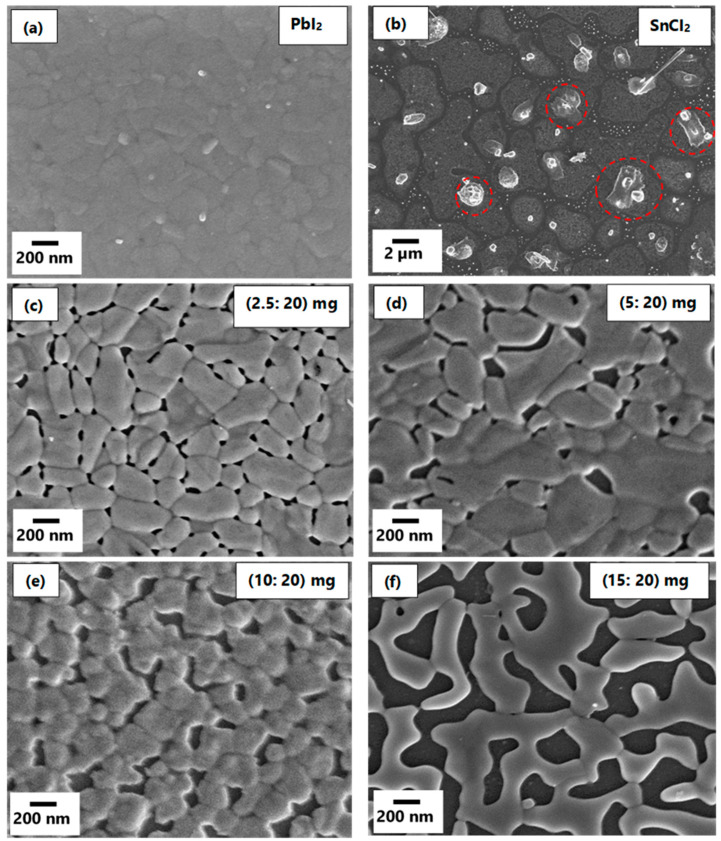
SEM micrographs of (**a**) PbI_2_ film, (**b**) SnCl_2_ films and (**c**–**f**) mixed metal -halide thin films.

**Figure 2 materials-14-03526-f002:**
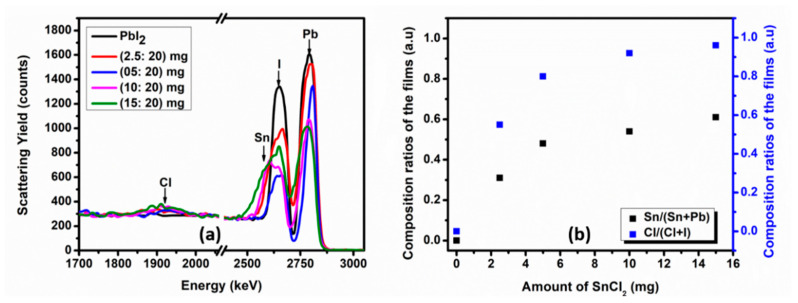
(**a**) The experimental RBS spectra of PbI_2_ and mixed SnCl_2_-PbI_2_ compound thin films, and (**b**) the compositional ratios of Sn/(Sn + Pb) and Cl/(Cl + I) in the precursor films.

**Figure 3 materials-14-03526-f003:**
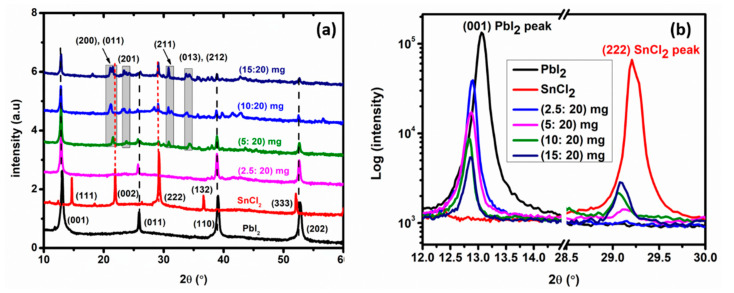
(**a**) X-ray diffraction pattern of PbI_2_, SnCl_2_ and mixed SnCl_2_-PbI_2_ compound film, (**b**) is the zoom-in to the (001) and (222) peaks.

**Figure 4 materials-14-03526-f004:**
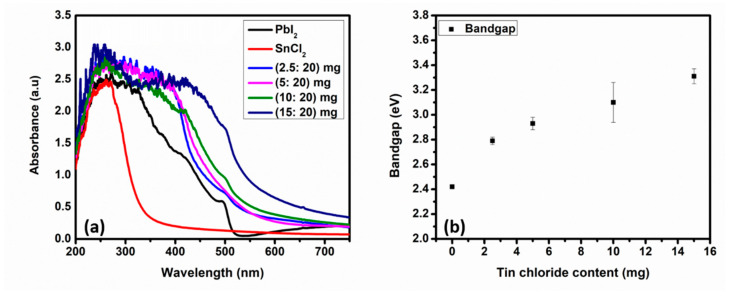
(**a**) The UV-Vis absorption spectra and (**b**) the E_g_ evolution plot upon addition of the different amount of SnCl_2_.

**Figure 5 materials-14-03526-f005:**
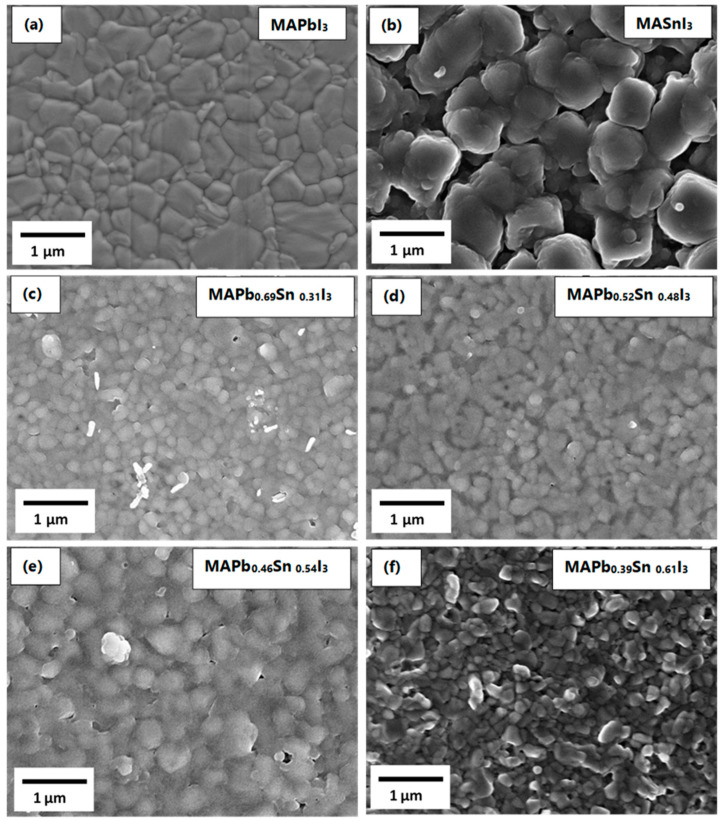
SEM micrographs of (**a**) MAPbI_3_ film, (**b**) MASnI_3_ films and (**c**–**f**) mixed metal-perovskites thin films.

**Figure 6 materials-14-03526-f006:**
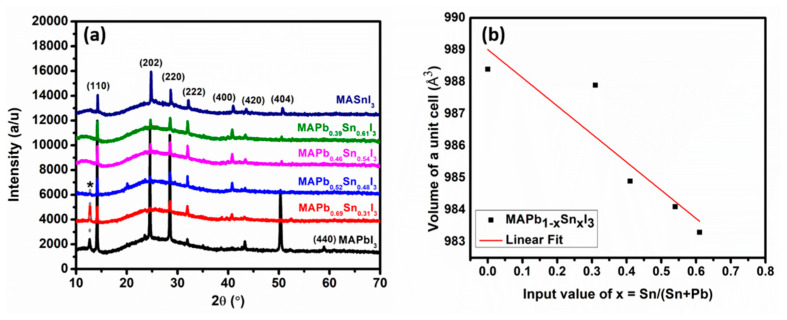
(**a**) X-ray diffraction pattern and (**b**) the evolution of the volume of the unit cells upon increasing content Sn. The linear fit is used to give direction to the eye.

**Figure 7 materials-14-03526-f007:**
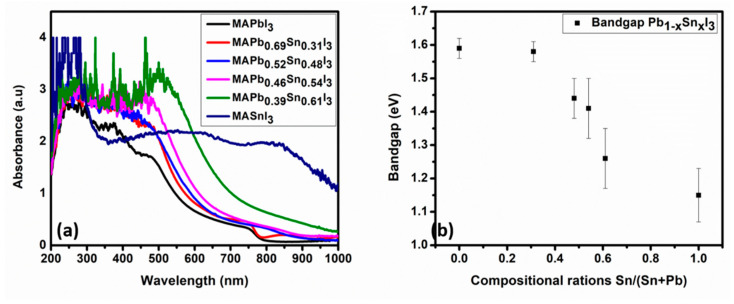
(**a**) UV-Vis absorption spectra of the perovskite films and (**b**) the E_g_ evolution plot for different compositional ratios of Sn. The direct allowed bandgap transition values were obtained by extrapolating the linear portion of the (*αhv*)^2^ versus *hv* plot, as depicted in Appendix A.

**Table 1 materials-14-03526-t001:** Stoichiometry obtained from the simulated RBS spectra.

Sample ID	Layer Depth (×10^17^ Atoms/cm^2^)		Cl	Sn	I	Pb
Lead iodide	L1	1.94	-	-	0.65	0.35
	L2	1.67	-	-	0.64	0.36
(2.5:20) mg	L1	2.28	0.18	0.07	0.44	0.31
	L2	1.94	0.54	0.14	0.16	0.10
(5:20) mg	L1	2.94	0.42	0.04	0.20	0.34
	L2	0.95	0.36	0.31	0.03	0.29
	L3	0.70	0.52	0.29	0.11	0.07
(10:20) mg	L1	1.40	0.48	0.15	0.03	0.34
	L2	1.51	0.38	0.33	0.05	0.24
	L3	2.03	0.50	0.34	0.04	0.12
(15:20) mg	L1	2.25	0.41	0.21	0.04	0.34
	L2	1.81	0.47	0.34	0.02	0.17
	L3	1.95	0.57	0.36	0.01	0.06

**Table 2 materials-14-03526-t002:** Lattice constants, FWHM of (110) peak and crystallite size of perovskite films.

Sample	a = b (Å)	c (Å)	Vol (Å^3^)	FWHM (ᵒ)	Crystallite Size (nm)
MAPbI_3_	8.85	12.62	988.4	0.1139	71.9
MAPb_0.69_Sn_0.31_I_3_	8.83	12.67	987.9	0.1788	45.8
MAPb_0.52_Sn_0.41_I_3_	8.82	12.66	984.9	0.1327	61.6
MAPb_0.46_Sn_0.54_I_3_	8.82	12.65	984.1	0.1568	52.2
MAPb_0.39_Sn_0.61_I_3_	8.82	12.64	983.3	0.1608	50.9
MASnI_3_	8.80	12.53	970.3	0.1254	65.6

**Table 3 materials-14-03526-t003:** Energy Bandgap and Urbach energy of binary metal perovskites.

Sample	Energy Bandgap (eV)	Urbach Energy (meV)
MAPbI_3_	1.59 ± 0.03	70 ± 10
MAPb_0.69_Sn_0.31_I_3_	1.58 ± 0.03	60 ± 10
MAPb_0.52_Sn_0.41_I_3_	1.44 ± 0.06	220 ± 10
MAPb_0.46_Sn_0.54_I_3_	1.41 ± 0.09	260 ± 10
MAPb_0.39_Sn_0.61_I_3_	1.26 ± 0.09	280 ± 60
MASnI_3_	1.15 ± 0.08	530 ± 60

## Data Availability

The data that support the findings of this study are available from the corresponding author, upon reasonable request.

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
