# Peer review of "Chemical Vapor Deposited Mixed Metal Halide Perovskite Thin Films"

_materials, 2021, doi:10.3390/ma14133526_

Round 1

Reviewer 1 Report

The figures could have better quality. Especially the graphs seem pixeled.

Author Response

Point 1: The figures could have better quality. Especially the graphs seem pixeled.

Response 1: Thank you for this comment. The quality of all the figures has been improved.

Reviewer 2 Report

The authors describe how CVD can be used to prepare thin films of mixed lead-tin perovskite using a two step procedure. They manage to control within a certain degree of accuracy the stoichiometry of the perovskite.

The amount of data is considerable even if sometimes they can be a little confusing and the message is not always clear. A general revision of the text could improve the clarity of the manuscript.

Furthermore, the main issues that the authors should comment before the manuscript could be considered for publication are:

1) the authors say that "This suggests that our film might have reacted with moisture to form impurities such as tin oxide or hydrogen chloride since their films were prepared and studied in the system without exposure to ambient air conditions."; considering that the fast oxidation of Sn(II) to Sn(IV) is the main reason of the instability of tin-based perovskites, have the authors tried to quantify the amount of oxygen absorbed/reacted with SnCl2 befor the evaporation as this aspect can dramatically change the properties of the final perovskite?

2) In figure 7a the UV-vis absorption spectra does not shoe a clear absorption edge suggesting a very high concentration of defects in the film; can the authors comment on this point?

3) figure 7b shows the dependance of the bandgap as a function of the Sn content in the film; the measured bandgaps in this case does not show the "anomalous bandgap" feature typical of the Sn-Pb mixed perovskites; moreover the bandgap for the all tin perovskite is very low (1.1eV), this value is very different compared to the usual values reported in literature; can the authors comment on this point?

Author Response

Point 1: The amount of data is considerable even if sometimes they can be a little confusing and the message is not always clear. A general revision of the text could improve the clarity of the manuscript.

Response 1: The text of the manuscript has been revised to improve the clarity of the manuscript.

Point 2: the authors say that "This suggests that our film might have reacted with moisture to form impurities such as tin oxide or hydrogen chloride since their films were prepared and studied in the system without exposure to ambient air conditions."; considering that the fast oxidation of Sn(II) to Sn(IV) is the main reason of the instability of tin-based perovskites, have the authors tried to quantify the amount of oxygen absorbed/reacted with SnCl2 befor the evaporation as this aspect can dramatically change the properties of the final perovskite?

Response 2: We could observe the reaction taking place immediately after removing the samples from the deposition chamber, with the XRD results discussed on page 7 of 19 confirmed that the SnCl2 film reacted with moisture to form a hydrated tin chloride thin film. We did not quantity the amount of oxygen absorbed by SnCl2 before the evaporation process, as it is expected that the initial level of oxidation would be similar for all samples. The Urbach energies have now been extracted for the perovskite thin films to quantify the defect density in the band gap that is associated with the level of oxidation (see point 3 below).

Point 3: In figure 7a the UV-vis absorption spectra does not shoe a clear absorption edge suggesting a very high concentration of defects in the film; can the authors comment on this point?

Response 3: Thank you for this comment and we agree. To quantify the defect concentration, particularly related to the band tails in the band gap, the Urbach energies were calculated from the UV/VIS data (Fig S6 and Table 3), which confirms an increase in defect density with increasing Sn-content in the perovskite thin films that is associated with the increased oxidation of Sn2+ to Sn4+ (manuscript updated on page 12 of 19).

Point 4: figure 7b shows the dependance of the bandgap as a function of the Sn content in the film; the measured bandgaps in this case does not show the "anomalous bandgap" feature typical of the Sn-Pb mixed perovskites; moreover the bandgap for the all tin perovskite is very low (1.1eV), this value is very different compared to the usual values reported in literature; can the authors comment on this point?

Response 4: Thank you for this comment. The 61% maximum Sn-content employed in this study is below the band gap bowing threshold, which is also noted in the revised manuscript on page 12 of 19.

The defect density in the band gap (Urbach energies, as in point 3 above) have now also been included in the discussion to provide insight into the difference in the band gap values of our films to those in literature.

Reviewer 3 Report

Although this article is of certain interest, nevertheless, it can be published only after clarification of some unclear and controversial points. In the part where optical properties are considered, the effects associated with the Urbach rule are completely ignored.  Correct allowance for the Urbach tail of the optical absorption and the corresponding exciton effects will allow you to more accurately define values of band-gap energies and all related ones.  It will be good if these values will be presented as a Table.  As an example, of Urbach tail in PbI2 and SnI2, see:

1. Sun, H., Zhu, X., Yang, D., Yang, J., Gao, X., & Li, X. (2014). Morphological and structural evolution during thermally physical vapor phase growth of PbI2 polycrystalline thin films. Journal of crystal growth405, 29-34.

 https://doi.org/10.1016/j.jcrysgro.2014.07.043

2. Hassoon, K. I., Mohammed, M. S., & Salman, G. D. (2019). Characterization of PbI2 thin films prepared by fast vacuum thermal evaporation. vol14, 401-406.  https://chalcogen.ro/401_HassoonKI.pdf

3. Ahmad, A., Saq’an, S., Lahlouh, B., Hassan, M., Alsaad, A., & El-Nasser, H. (2009). Ellipsometric characterization of PbI2 thin film on glass. Physica B: Condensed Matter404(1), 1-6.

https://doi.org/10.1016/j.physb.2008.09.041

4. Takeda, J., Ishihara, T., & Goto, T. (1985). Low energy tail of the exciton luminescence band in 2H-PbI2 and its relation to Urbach rule. Solid state communications56(1), 101-103.

https://doi.org/10.1016/0038-1098(85)90543-5

and etc

There is another important question about the stability and aging of the  PbSn-halides.  It is known, for example, that SnCl2 when standing in air, it gradually hydrolyzes with moisture.  

.

Author Response

Point 1: Although this article is of certain interest, nevertheless, it can be published only after clarification of some unclear and controversial points. In the part where optical properties are considered, the effects associated with the Urbach rule are completely ignored.  Correct allowance for the Urbach tail of the optical absorption and the corresponding exciton effects will allow you to more accurately define values of band-gap energies and all related ones.  It will be good if these values will be presented as a Table.  As an example, of Urbach tail in PbI2 and SnI2, see:

Response 1: Thank you for this important recommendation. The manuscript has now been updated to include the contributions of the defect states in the band gap, i.e. the Urbach energy. These values were approximated in figure S6 and are tabulated in table 3 on page 13 of 19. We appreciate this valuable insight and contribution.

Point 2: There is another important question about the stability and aging of the PbSn-halides.  It is known, for example, that SnCl2 when standing in air, it gradually hydrolyzes with moisture.

Response 2: The exposure of the precursor films to ambient environment was minimized due to this point that you are raising. This is noted in the manuscript on page 7 of 19 that the films formed are not pure SnCl2, rather a hydrated tin chloride (i.e., [SnCl4(H2O)2]3H2O) film with a monoclinic structure. Notwithstanding, literature suggest that that chlorine results in more stable perovskite thin films than iodine as part of the starting precursor films.

Round 2

Reviewer 2 Report

The authors have properly corrected the paper and it can be accepted in the current form.

Reviewer 3 Report

I am satisfied with the answers, so the article can be published now